# Infectious viral shedding of SARS-CoV-2 Delta following vaccination: A longitudinal cohort study

Miguel Garcia-Knight[1]☯, Khamal Anglin[2,3]☯, Michel Tassetto[1]☯, Scott Lu[2,3], Amethyst Zhang[1], Sarah A. Goldberg[2,3], Adam Catching[1], Michelle C. Davidson[4], Joshua R. Shak[4,5], Mariela Romero[2,3], Jesus Pineda-Ramirez[2,3], Ruth Diaz-Sanchez[2,3], Paulina Rugart[2,3], Kevin Donohue[4], Jonathan Massachi[4], Hannah M. Sans[4], Manuella Djomaleu[4], Sujata Mathur[2,3], Venice Servellita[6], David McIlwain[7], Brice Gaudiliere[7], Jessica Chen[2,3], Enrique O. Martinez[2,3], Jacqueline M. Tavs[2,3], Grace Bronstone[2,3], Jacob Weiss[2,3], John T. Watson[8], Melissa Briggs-Hagen[8], Glen R. Abedi[8], George W. Rutherford[2,3], Steven G. Deeks[9], Charles Chiu[6], Sharon Saydah[8], Michael J. Peluso[9], Claire M. Midgley[8], Jeffrey N. Martin[2], Raul Andino[1]*, J. Daniel Kelly[2,3,5,10]*

**1** Department of Microbiology and Immunology, UCSF, California, United States of America, **2** Department of Epidemiology and Biostatistics, University of California, San Francisco, California, United States of America, **3** Institute for Global Health Sciences, University of California, San Francisco, California, United States of America, **4** School of Medicine, University of California, San Francisco, California, United States of America, **5** San Francisco VA Medical Center, San Francisco, California, United States of America, **6** Division of Infectious Diseases, UCSF, California, United States of America, **7** Department of Microbiology and Immunology, Stanford, California, United States of America, **8** Respiratory Viruses Branch, Division of Viral Diseases, CDC, Atlanta, Georgia, United States of America, **9** Division of HIV, Infectious Disease, and Global Medicine, UCSF, California, United States of America, **10** F.I. Proctor Foundation, University of California, San Francisco, California, United States of America

☯ These authors contributed equally to this work.
* raul.andino@ucsf.edu (RA); dan.kelly@ucsf.edu (JDK)

**Data Availability Statement:** All relevant data are within the manuscript and its Supporting Information files.

## Abstract

The impact of vaccination on SARS-CoV-2 infectiousness is not well understood. We compared longitudinal viral shedding dynamics in unvaccinated and fully vaccinated adults. SARS-CoV-2-infected adults were enrolled within 5 days of symptom onset and nasal specimens were self-collected daily for two weeks and intermittently for an additional two weeks. SARS-CoV-2 RNA load and infectious virus were analyzed relative to symptom onset stratified by vaccination status. We tested 1080 nasal specimens from 52 unvaccinated adults enrolled in the pre-Delta period and 32 fully vaccinated adults with predominantly Delta infections. While we observed no differences by vaccination status in maximum RNA levels, maximum infectious titers and the median duration of viral RNA shedding, the rate of decay from the maximum RNA load was faster among vaccinated; maximum infectious titers and maximum RNA levels were highly correlated. Furthermore, amongst participants with infectious virus, median duration of infectious virus detection was reduced from 7.5 days (IQR: 6.0–9.0) in unvaccinated participants to 6 days (IQR: 5.0–8.0) in those vaccinated (P = 0.02). Accordingly, the odds of shedding infectious virus from days 6 to 12 post-onset were lower among vaccinated participants than unvaccinated participants (OR 0.42 95% CI 0.19–

**Funding:** The FindCOVID study at UCSF is funded by the Centers for Disease Control and Prevention (CDC Contract FE_08009). JDK was supported during this study by the National Institute of Allergy and Infectious Diseases (K23 grant number AI146268). The funders had no role in study design, data collection and analysis, decision to publish, or preparation of the manuscript.

**Competing interests:** The authors have declared that no competing interests exist.

0.89). These results indicate that vaccination had reduced the probability of shedding infectious virus after 5 days from symptom onset.

## Author summary

We present longitudinal data on the magnitude, duration and decay rate of viral RNA and the magnitude and duration of infectious virus in nasal specimens from vaccinated and unvaccinated participants. On average, vaccinated participants (infected with the highly transmissible Delta variant) showed a lower probability of having infectious virus after 5 days of symptoms compared to unvaccinated participants (infected with mostly pre-delta viral lineages), even though both groups had a similar magnitude of infectious virus at or near the peak. These data help improve our understanding of the duration of the infectious period when infection occurs following vaccination and serves as a reference for future studies of shedding dynamics following infections with novel variants of concern.

## Introduction

Understanding the impact of vaccination on viral shedding and infectiousness of SARS-CoV-2 is key to the public health response against the COVID-19 pandemic. COVID-19 vaccines are highly effective against severe COVID-19 illness including hospitalization and death [1] and reduce transmission [2–4]. However, effectiveness against infection is variable, in part due to the emergence of viral variants of concern (VOC) which are able to evade neutralizing antibody responses [5]. Clinical studies have shown a reduction in vaccine effectiveness against symptomatic infection with the VOCs Delta [6,7], Beta [8], Gamma [9] and Omicron [10,11]. Prior to December 2021, the Delta and Gamma variants were associated with a higher proportion of infections in the United States (US) following vaccination compared to other variants that were dominant [12,13] and vaccine breakthrough infections with these variants were documented in diverse settings [14,15]. How vaccination modulates viral shedding and the infectious period of highly transmissible viral variants is not well understood.

Most investigations of SARS-CoV-2 shedding by vaccination status have focused on viral RNA [16–20]. Comparisons of vaccinated and unvaccinated individuals show limited effects of vaccination on peak viral RNA loads [19,20], particularly as time since vaccination increases [17]. By contrast, faster viral RNA clearance has been shown following vaccination [18–20], suggesting a potential shortening of the infectious period. However, as viral RNA is not a direct measure of infectious virus and viral RNA is detected for substantially longer than infectious virus among unvaccinated individuals [21], a critical gap exists when inferring potential periods of infectiousness following vaccination.

To address this, we analyzed specimens from SARS-CoV-2-infected adults who formed part of a longitudinal cohort established to characterize household transmission dynamics and the natural history of SARS-CoV-2 infection among non-hospitalized persons. Participants were recruited from September 2020 to October 2021, a period that spanned successive waves of the pandemic, and the introduction of vaccines against COVID-19. We compared key shedding outcomes between participants stratified by vaccination status.

## Results

### Baseline characteristics

Our analysis included 84 non-hospitalized participants with acute SARS-CoV-2 infection (based on health provider-ordered molecular testing) who were ≥18 years of age, immunocompetent, and did not receive monoclonal antibody therapy (**Fig 1**). Of these, 52 (62%) were unvaccinated and 32 (38%) had received a full primary series of a COVID-19 vaccine under EUA by FDA. While study enrollment was continuous, unvaccinated adults were enrolled during September 2020 –September 2021 and vaccinated adults were enrolled during April–October 2021. Participants were mainly vaccinated with mRNA vaccines (94%) and were enrolled a median of 142 (IQR 88–168) days following the second dose. Age, sex, race/ethnicity and underlying conditions were similar between groups (**Table 1**). A larger proportion of unvaccinated participants had a BMI ≥30 compared to vaccinated ones. A majority of both unvaccinated (49/52 [94%]) and vaccinated participants reported having a symptomatic infection (32/32 [100%]). We sequenced 37/52 (71%) and 26/32 (81%) viral genomes from unvaccinated and vaccinated individuals, respectively. In accordance with the circulation of viral lineages at the time of recruitment, among unvaccinated individuals, 30/52 (58%) were infected with lineages not classified as variants of concern and 5/52(10%) and 1/52 (2%) were Epsilon and Alpha lineage, respectively. By contrast, 26/32 (81%) infections following vaccination were Delta lineage and sub-lineages. IgG responses against N were assessed in 54% of participants at enrolment and no responses were detected, suggesting infections were primary infections.

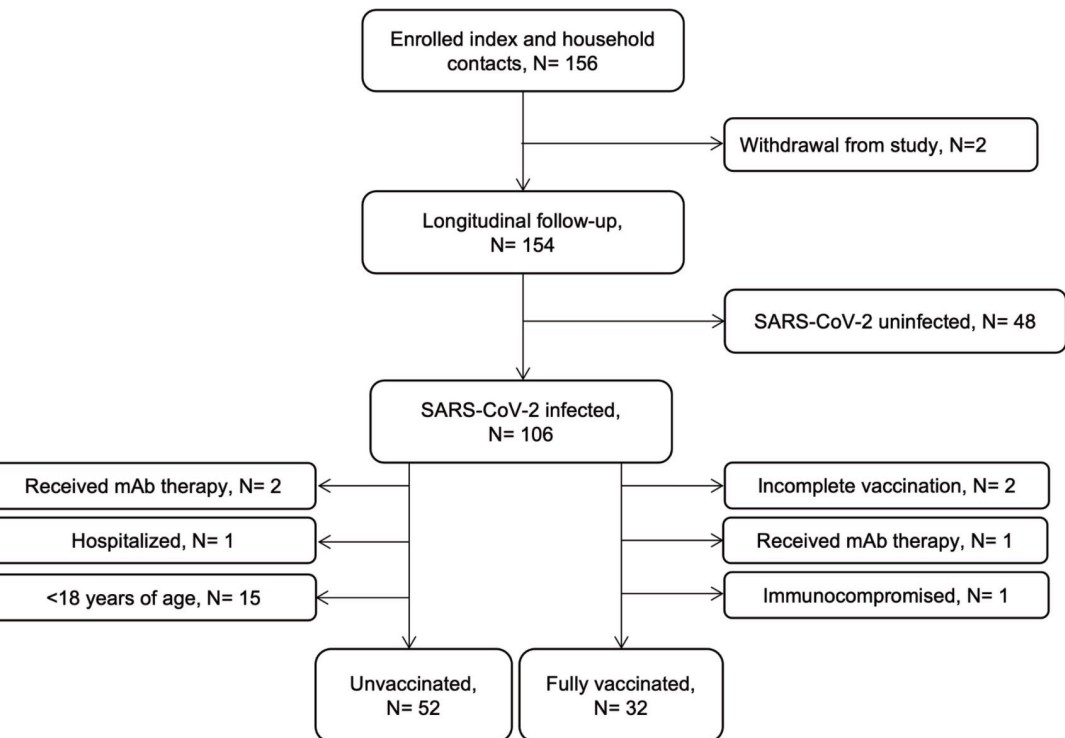

**Fig 1. Flow diagram of patient inclusion.** Number of participants included in the study analysis. Excluded participants are indicated in lateral text boxes.

**Table 1. Participant demographics, vaccination status and viral characteristics.**

| Characteristic | Number of participants (%)[a] | | | P value[b] |
| --- | --- | --- | --- | --- |
| | Unvaccinated (N = 52) | Vaccinated (N = 32) | Total (N = 84) | |
| **Age group, yrs** | | | | |
| 18–29 | 13 (25) | 9 (28) | 22 (26) | |
| 30–39 | 15 (29) | 13 (41) | 28 (33) | |
| 40–49 | 14 (27) | 8 (25) | 22 (26) | |
| 50–59 | 5 (10) | 0 (0.0) | 5 (6) | |
| ≥60 | 5 (10) | 2 (6.0) | 7 (8) | |
| Median (IQR) | 38.5 (29.5–48.0) | 37.0 (28.0–40.0) | 37.0 (29.0–44.3) | 0.24 |
| **Sex** | | | | |
| Female | 28 (54) | 19 (59) | 47 (56) | 0.69 |
| **Race/Ethnicity** | | | | |
| Hispanic/Latino | 13 (25) | 4 (13) | 17 (20) | 0.54 |
| White | 21 (40) | 17 (53) | 38 (45) | |
| Black/African American | 2 (4) | 1 (3) | 3 (4) | |
| Asian | 10 (19) | 3 (9) | 13 (16) | |
| Pacific Islander/Native Hawaiian | 2 (4) | 0 (0) | 2 (2) | |
| Native American or Alaska Native | 1 (2.0) | 1 (3.0) | 2 (2) | |
| Unknown | 3 (6) | 6 (19) | 9 (11) | |
| **BMI group** | | | | |
| ≤24.9 | 16 (31) | 18 (56) | 34 (40) | 0.02 |
| 25 to 29.9 | 17 (33) | 8 (25) | 25 (29) | |
| ≥30 | 15 (29) | 2 (6) | 15 (20) | |
| Unknown | 4 (8) | 4 (13) | 5 (11) | |
| **Underlying conditions[c]** | | | | |
| Autoimmune disease | 1 (2) | 0 (0) | 1 (1) | NA |
| History of cancer | 2 (4) | 0 (0) | 2 (3) | NA |
| Diabetes | 2 (4) | 0 (0) | 2 (3) | NA |
| Hypertension or high blood pressure | 4 (8) | 3 (9) | 7 (8) | 0.69 |
| Lung pathology[d] | 10 (20) | 5 (24) | 15 (21) | 1.0 |
| **Vaccine** | | | | |
| BNT162b2 (Pfizer) | NA | 19 (59) | 19 (23) | NA |
| mRNA-1273 (Moderna) | NA | 11 (34) | 11 (13) | NA |
| JNJ-78436735 (J&J) | NA | 2 (6) | 2 (2) | NA |
| Days since vaccination, median (IQR) | NA | 142 (88–168) | | NA |
| >14 days to <4 months[e] | NA | 10 (31) | | NA |
| ≥4 months[e] | NA | 22 (69) | | NA |
| **Viral lineage** | | | | |
| Delta (B.1.617.2)[f] | 0 (0.0) | 26 (81) | 26 (31) | <0.01 |
| Epsilon (B.1.427/9) | 5 (10) | 0 (0.0) | 5 (6) | |
| Alpha (B.1.1.7) | 1 (2) | 0 (0.0) | 1 (1) | |
| Non-VOI/VOC | 30 (58) | 0 (0.0) | 30 (36) | |
| Unknown[g] | 16 (31) | 6 (19) | 22 (26) | |

(*Continued*)

**Table 1.** (*Continued*)

| Characteristic | Number of participants (%)[a] | | | P value[b] |
|---|---|---|---|---|
| | Unvaccinated (N = 52) | Vaccinated (N = 32) | Total (N = 84) | |
| Anti N IgG at enrolment[h] | 0 (0) | 0 (0) | 0 (0) | NA |

[a]Unless indicated.

[b]Chi squared test.

[c]Self-reported, no cases of HIV/AIDS, heart attack or heart failure or kidney disease reported.

[d]e.g., asthma, cryptogenic organizing pneumonia.

[e]From second dose.

[f]Sub-lineages included AY.1, AY.3, AY.4, AY.15 & AY.25.

[g]Sequence not determined due to high Ct values.

[h]14/52 and 31/32 unvaccinated and vaccinated tested, respectively. BMI, body mass index, $kg/m^2$; VOI, variant of interest; VOC, variant of concern. IQR, interquartile range.

## Viral RNA target abundance and shedding dynamics

The quantity of viral RNA was assessed in a total of 1080 nasal specimens from 84 participants (S1 Fig) stored under RNA-preserving conditions (S2 Fig). Overall, copies of nucleocapsid (N) and envelope (E) genes in each nasal specimen were of similar magnitudes and highly correlated with each other ($R = 0.98$, P <0.001) (S3 Fig). A median of 13.0 (IQR: 12.0–14.0) and 14.0 (IQR: 13.0–14.5) samples were collected for each unvaccinated and vaccinated participant, respectively (P = 0.25; Table 2). A total of six (11%) unvaccinated and five (16%) vaccinated participants did not have detectable RNA in any nasal sample (Table 2) despite being SARS-CoV-2 RNA-positive by health provider-ordered clinical testing prior to enrollment. The number of RNA positive samples per participant, defined via amplification of both N and E, did not differ between groups (P = 0.76).

**Table 2. Viral shedding outcomes for adult participants, by vaccination status.**

| Outcome | Unvaccinated (N = 52) | Fully vaccinated (N = 32) | P value[a] |
|---|---|---|---|
| Number of specimens tested by RT-qPCR per participant, median (IQR) | 13.0 (12.0–14.0) | 14.0 (13.0–14.5) | 0.25 |
| Participants with ≥1 nasal specimen positive for viral RNA, no. (%) | 46 (89) | 27 (84) | 0.76[b] |
| Number of specimens positive for both N and E per participant, median (IQR) | 5.0 (1.5–8.0) | 5.0 (3.0–7.5) | 0.55 |
| Maximum N copies/mL, $x10^7$, median (IQR) | 7.5 (0.004–108.0) | 1.0 (0.2–32.0) | 0.39 |
| Maximum E copies/mL, $x10^7$, median (IQR) | 1.2 (0.005–27.4) | 0.6 (0.07–23.3) | 0.88 |
| Duration of viral RNA shedding, median days (IQR) | 9.0 (7.0–12.0) | 8.0 (4.5–12.0) | 0.52 |
| Participants with ≥1 nasal specimen positive for viral culture, no (%) | 38 (73) | 23 (72) | 1.00[b] |
| Number of specimens with successful culture of infectious virus per participant, median (IQR) | 3.5 (0.0–6.0) | 2.0 (0.0–4.5) | 0.08 |
| Duration of infectious viral shedding among participants with ≥1 nasal sample positive for viral culture, median days (IQR)[c] | 7.5 (6.0–9.0) | 6.0 (5.0–8.0) | 0.02 |
| Duration of infectious viral shedding among all participants, median days (IQR) | 7 (0–8) | 5 (0–7) | 0.12 |
| Maximum infectious titer (PFU/mL), $x10^3$, among all participants, median (IQR) | 8.4 (0–240) | 1.6 (0–1160) | 0.60 |
| Participants with infectious virus detected >5 days post symptom onset, no. (%) | 32 (62) | 12 (38) | 0.03 |
| Participants with infectious virus detected >10 days post symptom onset, no. (%) | 5 (10) | 1 (3) | 0.41 |

[a]Mann-Whitney U test.

[b]Fisher's Exact.

[c]Analysis on N = 38 unvaccinated and N = 23 vaccinated participants. N, nucleocapsid gene; E, envelope gene; IQR, interquartile range; PFU, plaque forming units.

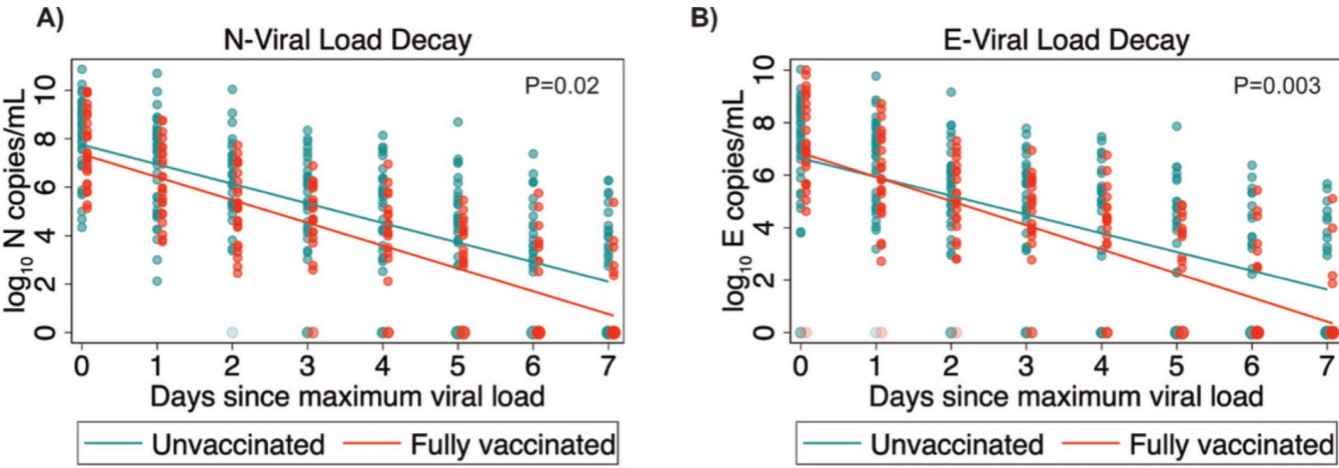

**Fig 2. Dynamics of viral RNA decay, by vaccination status.** Mixed effects model predicting mean viral RNA load of N (A) and E (B) targets longitudinally in unvaccinated and vaccinated participants. The analysis was done from the maximum detected viral RNA load of each participant (day 0) over seven days. Linear slopes were fitted to the data from vaccinated and unvaccinated groups and the P value for the significance of the interaction between vaccination status and time is shown. The color gradient of dots along the x axis represents increased proportions of specimens with no detectable viral RNA.

We compared the magnitude, duration, and decay rates of viral RNA targets between unvaccinated and vaccinated participants (Table 2). No differences were found between groups in the median maximum copies of N ($7.5 \times 10^7$ [IQR: $4.0 \times 10^4$-$1.1 \times 10^9$] vs $1.0 \times 10^7$ [IQR: $0.2 \times 10^{7.}$ - $3.2 \times 10^8$] log10 copies/mL, P = 0.39) or E ($1.2 \times 10^7$ [IQR: $5.0 \times 10^4$–$2.7 \times 10^8$] vs $0.6 \times 10^7$ [IQR: $7.0 \times 10^5$–$2.3 \times 10^8$] log10 copies/mL; P = 0.88). Similarly, the duration of detection of dual N and E positive specimens (9.0 [IQR: 7.0–12.0] vs 8.0 [IQR: 4.5–12.0], P = 0.52) did not differ between unvaccinated and vaccinated participants (Table 2). However, we observed a significantly faster rate of viral RNA load decline for N (P = 0.02) and E (P = 0.003) for 7 days after maximum viral load among vaccinated participants compared to unvaccinated participants (Fig 2).

## Infectious viral shedding dynamics

To define the infectious period at the individual level, viral culture was done in a targeted manner on 416/650 (64%) specimens from unvaccinated participants and 242/430 (56%) specimens from vaccinated participants (S1 Fig). In addition, infectious titers in specimens from each participant with the maximum quantity of viral RNA were determined. Specimens were stored under conditions that preserve infectious virus titres (S2 Fig). The proportion of unvaccinated (38/52 [73%]) and vaccinated (23/32 [72%]) participants with ≥1 specimens with detectable infectious virus did not differ (P = 1.0); however, vaccinated adults tended to have fewer samples with infectious virus detected (P = 0.08; Table 2). Amongst participants with ≥1 specimen with detectable infectious virus, the median duration of infectious viral shedding was significantly shorter in vaccinated versus unvaccinated individuals (median of 6.0 days (IQR: 5.0–8.0) vs 7.5 days (IQR: 6.0–9.0) (P = 0.02; Table 2). However, this comparison was not statistically significant when including participants who had no detectable infectious virus in any sample (P = 0.12). The duration of infectious virus shedding was not found to be modulated by mRNA vaccine type or time between vaccination and infection (S2 Table).

Overall, the duration of infectious virus shedding was significantly correlated with the maximum viral RNA load (P <0.01; S4 Fig). In addition, maximum viral RNA load was strongly correlated with the infectious viral titer in the same specimen ($R^2$ = 0.78, P = <0.001; Fig 3A).

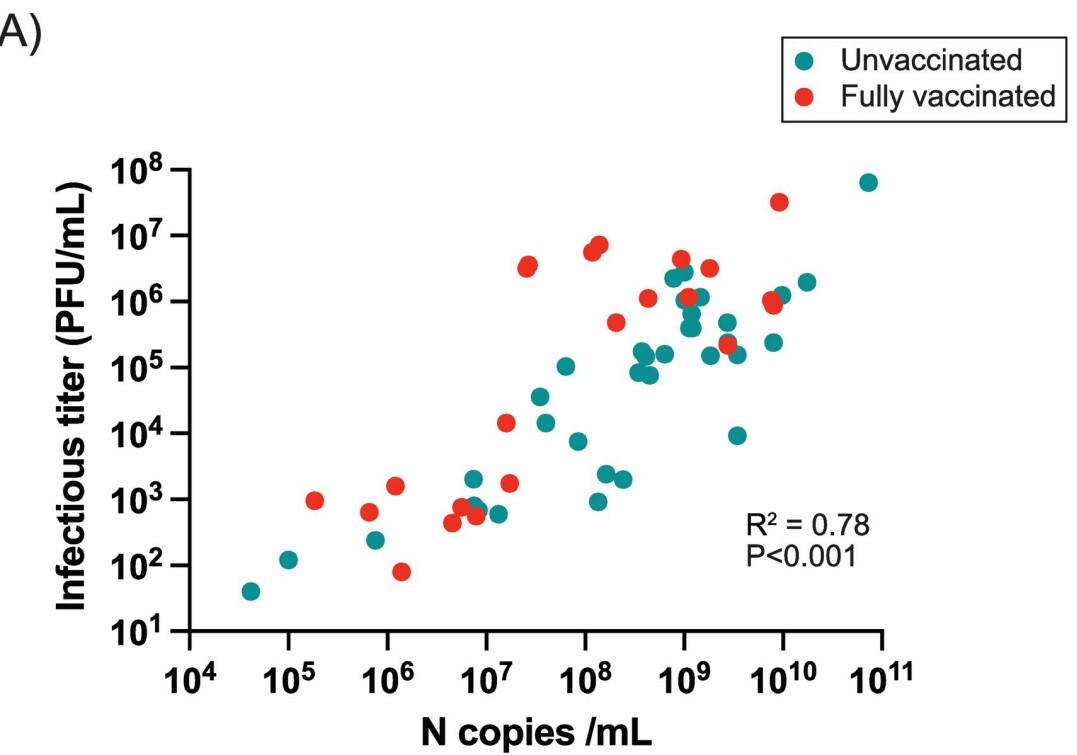

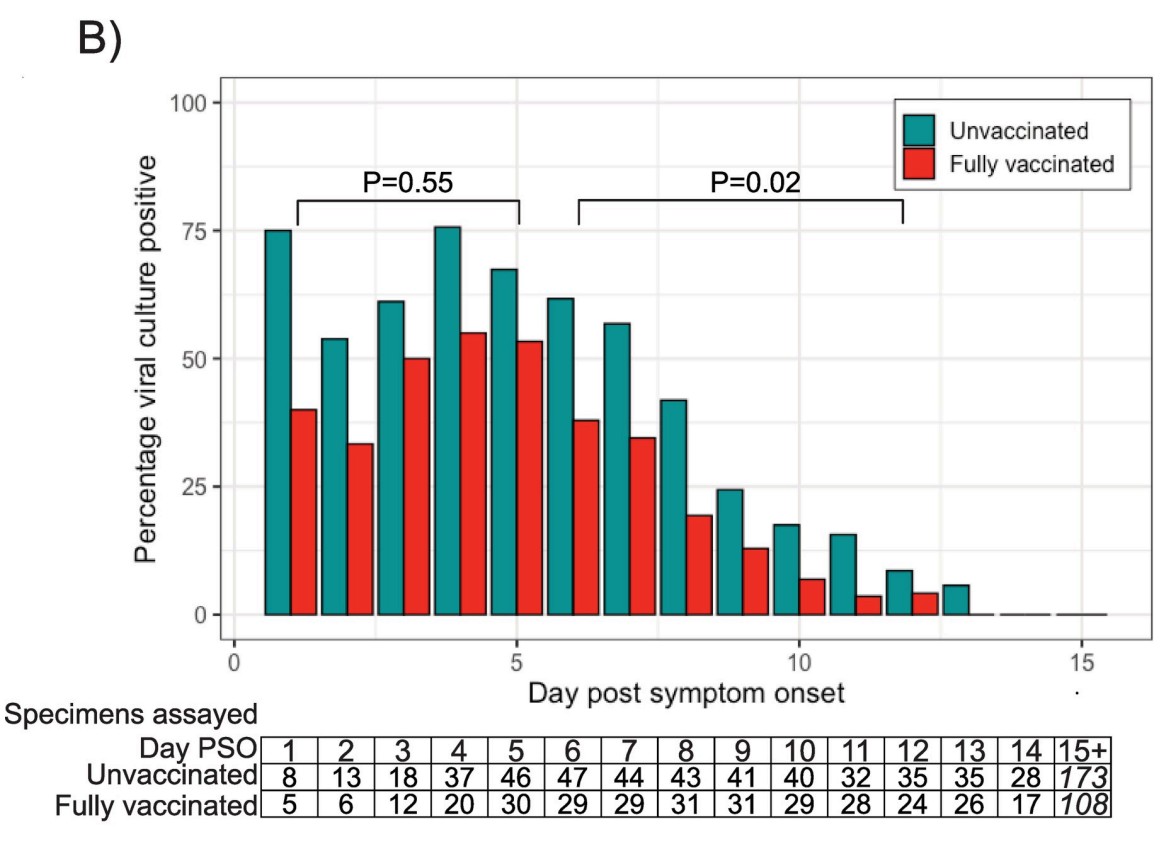

**Fig 3. Proportion of specimens with infectious virus over the infectious period.** Correlation between the maximum viral RNA level (N copies/ml) for a given participant and the infectious viral titer of that same nasal specimen. Pearson's correlation was used to assess the strength of the correlation. B) The proportion of nasal samples with infectious virus in unvaccinated and vaccinated participants by day post symptom onset. The number of specimens assessed at each time point is indicated in the lower box. The last column is a sum of all samples assessed from day 15 onwards. Logistic regression analysis spanning days 1–5 and 6–12 was done to assess statistically significant differences between groups.

No differences in maximum infectious titers were observed between vaccination groups (P = 0.60, **Table 2**).

Throughout the infectious period, the proportion of specimens with infectious virus was lower in fully vaccinated participants compared to unvaccinated participants (**Fig 3B**). The odds of detecting infectious virus over the first 5 days post symptom onset did not differ for vaccinated compared to unvaccinated participants (OR 0.76, 95% CI 0.29–1.95). However, from days 6–12, the odds were reduced for vaccinated compared to unvaccinated participants (OR 0.42 95% CI 0.19–0.89, P = 0.02). In addition, a lower proportion of vaccinated participants had infectious virus after 5 days of symptom onset compared to unvaccinated participants (**Table 2**, P = 0.03), though after day 10 this comparison was underpowered. Symptom resolution preceded the cessation of infectious viral shedding in most participants; however, amongst the 44 symptomatic participants with infectious virus shedding for > 5 days after symptom onset, 3/32 (9.4%) of unvaccinated and 0/12 (0%) of fully vaccinated participants had infectious virus after symptoms resolved.

## Discussion

Our analysis indicates that when infected with SARS-CoV-2, individuals with a full primary vaccination course have a reduced likelihood of shedding infectious virus from the nose from 5 days after symptom onset, have a more rapid decline of viral RNA from day of maximal load, and, when infectious, exhibit a shorter duration of infectious virus detection, as compared to unvaccinated individuals. However, we observed no impact on the magnitude of infectious virus at or near the peak of infectivity. These combined data suggest that although full vaccination may be inefficient at reducing infectiousness in early infection, it likely leads to a reduction in the duration of infectiousness. These findings are consistent with previous reports that fully vaccinated adults are less likely to transmit SARS-CoV-2 to others [20,22–25] and underscore the need for novel vaccination strategies that reduce viral titers during peak infectivity.

These findings help fill an important gap in our understanding of the infectious period of non-hospitalized adults with SARS-CoV-2 infections. While studies have addressed in detail the longitudinal dynamics of RNA shedding [18–20], indicating an accelerated decay of viral RNA over the acute infectious period among those vaccinated, less is known about the duration of infectious virus shedding. Studies analyzing the correlation between culture positivity and RT-qPCR cycle threshold (Ct) have suggested that vaccination leads to a reduced probability of culture positivity for a given RNA load [26,27]. However, a cross sectional comparison of quantitative infectious viral titers in samples matched by Ct found no differences between vaccinated and unvaccinated individuals [28]. Longitudinal analyses of infectious viral shedding comparing vaccinated against unvaccinated individuals have shown conflicting outcomes, depending on the study population. In studies focused on university students and employees [26,29], the infectious period was shorter in vaccinated individuals. By contrast, in a federal prison population [29] the infectious period was similar in vaccinated and unvaccinated persons. Recent data focused on the infectious virus titers over the first five days following symptom onset additionally indicate that vaccination, in the context of Delta circulation, leads to a reduction in the magnitude of infectious virus in early infection compared to unvaccinated

persons [30]. Our observation that maximum virus titer and RNA load are highly correlated, together with faster decline of viral RNA in vaccinated individuals, suggest that the rate of infectious decline may also be impacted by vaccination.

We found significant heterogeneity in both infectious virus and viral RNA shedding dynamics in both vaccinated and unvaccinated groups with the IQR of maximum viral RNA loads spanning 6 $\log_{10}$ scales. In addition, despite clinical RT-qPCR diagnosis, ~13% of all participants had no detectable viral RNA in any nasal sample and 27% and 28% of unvaccinated and vaccinated participants, respectively, had no infectious virus. This heterogeneity was independent of vaccination and is in line with findings from longitudinal shedding studies in various populations [20,31] and a recent human challenge study [32]. Previous infections in our cohort were unlikely to contribute to these observations as evidenced by a lack of IgG responses to N in all those participants tested at enrolment. Participants not screened for anti N responses were recruited early in the study when reinfection was rare phenomenon [33,34]. Therefore, the mechanisms that underpin heterogeneity of viral shedding of SARS-CoV-2, are unclear, may involve infectious dose, pre-existing adaptive immunity [35], host genetics [36] and innate immune responses [37], among others, and merit targeted studies.

We identified infectious viral shedding beyond day 10 from symptom onset among both unvaccinated and vaccinated non-hospitalized adults. Prior studies in mild-to-moderately ill individuals show concordant findings [38], whilst others did not detect infectious virus beyond 10 days after symptom onset [21,39]; a longer infectious period has been observed in severe infections [40]. These differences may be related largely to the methods used to detect infectious virus and cell lines. We used Vero TMPRSS2/hACE-2 cells that were engineered to prevent furin cleavage site mutations in spike following serial passage [41] and may provide enhanced sensitivity to detect infectious virus compared to other commonly used cell lines. To our knowledge, this is the first infectious viral shedding study to use cells co-expressing TMPRSS2/hACE-2. Host factors such age and comorbidities and infecting viral lineage may have also contributed to differences between studies. Despite epidemiological evidence that most transmission occurs early in the infectious period [42], our findings provide a biological rationale for why some degree of transmission still occurs in the late shedding period (beyond 5 but not after 10 days of symptoms) particularly in those with high peak viral loads (**S3 Fig**) [43]. Additional studies are needed to establish the thresholds of infectious virus titers in late infection required for community transmission and whether anamnestic neutralizing antibody responses may play a role limiting the infectious period even when viral loads remain high.

Our study has some limitations. Measurement bias from self-reported symptoms may have occurred from recall error and viral variant differences among participants. Differences in viral load may have had an effect on the day of symptom onset in participants relative to initial infection, as variant-specific changes in symptom outcomes have been reported [44]. Furthermore, unvaccinated individuals in our analysis were mostly enrolled prior to the emergence and global spread of the Delta lineage, whereas fully vaccinated individuals were almost exclusively enrolled once Delta was the dominant circulating lineage. Thus, we were unable to control for this as a potential confounder. Previous studies have indicated that infections with Delta lineage viruses have higher peak viral loads [30,45] and longer duration of shedding than pre-Delta lineages [46,47]. This suggests that we may have underestimated differences between vaccination groups, bringing our results from early infection in line with those from Puhach et al. (2022) [30]. In addition, index cases were from the list of SARS-CoV-2 positive cases at UCSF-affiliated medical centers in San Francisco and may not be representative of all SARS-CoV-2 infections occurring in the same jurisdiction. We were also underpowered to analyze shedding dynamics during the pre-symptomatic and early symptomatic period and thus we may have missed the peak viral RNA load in some individuals and thus underestimated

maximum values (**S1 Fig**). Lastly, our study was likely underpowered to detect differences in overall infectious virus shedding duration between groups, though our comparison when restricted to participants with infectious virus (**Table 2**) is in line with the regression analysis in **Fig 3**.

In summary, in addition to the protective effect of COVID-19 vaccines against severe disease, we provide evidence that full primary vaccination may reduce infectiousness during the late stage of acute infection. The impact of this finding on the transmission dynamics in households and in the community is a key question that needs further investigation. As the SARS-CoV-2 pandemic continues and the landscape of viral variants and vaccine formulations change, monitoring periods of infectiousness and their association with transmission will be key to developing public health policies that maintain transmission of SARS-CoV-2 at levels that limit morbidity, mortality, and societal disruption.

## Materials and methods

### Ethics statement

The study was reviewed by the UCSF Institutional Review Board and given a designation of public health surveillance according to federal regulations as summarized in 45 CFR 46.102(d) (1)(2). Approval number IRB# 20–30388. Written informed consent was obtained from all participants.

**Study design.** Participants were part of an observational longitudinal cohort based in the San Francisco Bay Area initiated in August 2020 designed to study shedding dynamics in early infection and transmission in a household setting. Index cases (IC) were identified from individuals with a positive health provider-ordered SARS-CoV-2 nucleic acid amplification test result on a nasopharyngeal or oropharyngeal (NP/OP) specimen done at UCSF-affiliated health facilities. ICs were recruited if they were within 5 days of symptom onset, had ≥1 household member (HM) and had no HMs with COVID-19 symptoms in the preceding week. Participants under 18 years of age and adults who were immunocompromised, hospitalized or who had received monoclonal antibody therapy, were excluded from this analysis. Participants were provided with specimen collection kits with instructions and self-collection demonstrations by study clinicians. For ICs, specimen collection was targeted for the day of enrollment (dE), daily thereafter up to day 14 and on days 17, 19, 21 and 28 post symptom onset. For HMs, daily specimen collection was targeted from dE and thereafter on the same days as the IC, counted relative to symptom onset of the IC. For all participants, symptom surveys, clinical and epidemiologic questionnaires, done either by phone or written, were completed on dE, and on days 9, 14, 21, and 28 after IC symptom onset. The type and timing of COVID-19 vaccination was verified with the vaccination card of each participant at enrollment and each subsequent survey. To be included in this analysis, vaccinated participants had to be fully vaccinated, defined as the receipt of all recommended doses (2 doses ≥28 days apart of an mRNA vaccine or 1 dose of JNJ-78436735) of a COVID-19 primary vaccine series under EUA by FDA, 14 days prior to infection or enrolment.

**Sample collection.** All anterior nasal specimens were self-collected by participants. We instructed participants to rotate flocked swabs 5 times in each nostril and place in conical tubes containing 3mL of Viral Transport Medium (CDC SOP# DSR-052-03). Samples were stored in study-dedicated -20˚C freezers at the participant's homes until weekly collection by the study team, at which time they were transported on dry ice for storage at -80˚C. For study assays, the samples were thawed and aliquoted into screw cap microtubes. RNA extraction was done after this initial thaw cycle and virus culture was attempted following a second freeze-

thaw cycle. We performed additional testing to evaluate for infectious viral degradation between these freeze-thaw cycles and did not observe any evidence of degradation.

**Detection of SARS-CoV-2 anti-N IgG.** Blood samples collected by the participants were taken to the UCSF Clinical Laboratory for SARS-CoV-2 anti-nucleocapsid (N) IgG testing using the Abbott assay.

**RNA extraction.** Automated RNA extraction was done using the KingFisher (Thermo Scientific) automated extraction instrument and compatible extraction kits in a 96 well format. 200uL of nasal samples were used for extraction with the MagMAX Viral/Pathogen Nucleic Acid Isolation Kit (Thermo Scientific) following the manufacturer's protocol and eluted into 50uL of nuclease free water. For confirmatory RT-qPCR following viral culture, RNA extraction was done using 300uL of inactivated sample (1:1 mixture of 2x RNA/DNA Shield and culture supernatant) and the Quick-DNA/RNA Viral MagBead kit (Zymo).

**RT-qPCR assay.** For each RT-qPCR reaction, 4μL of RNA sample were mixed with 5μL 2x Luna Universal Probe One-Step Reaction Mix, 0.5μL 20x WarmStart RT Enzyme Mix (NEB), 0.5μL of target gene specific forward and reverse primers and probe mix (**S1 Table**). RT-qPCR were run for SARS-CoV2 N and E and for host mRNA, *RNaseP*, as a control for RNA extraction. Primers (forward and reverse) and probe concentrations in each mix used per RT-qPCR reaction were as follows: 8μM forward/reverse each and 4μM probe for E, 5.6μM forward/reverse each and 1.4μM probe for N and 4μM forward/reverse each and 1μM probe for RNaseP. Each 96 well RT-qPCR plate was run with a 10-fold serial dilution of an equal mix of plasmids containing a full copy of nucleocapsid (N) and envelope (E) genes (IDT) as an absolute standard for RNA copies calculation and primer efficiency assessment. RT-qPCR were run on a CFX Connect Real-Time PCR detection system (Biorad) with the following settings: 55°C for 10 min, 95°C for 1 min, and then cycled 40 times at 95°C for 10s followed by 60°C for 30s. Probe fluorescence was measured at the end of each cycle. All probes, primers and standards were purchased from IDT. We defined a sample as being RNA positive if both N and E were detected at Ct ≤40. These two RNA targets were selected to optimize specificity of the PCR platform, excluding spurious RT-qPCR signals that would have been false positives. To control for the quality of self-sampling, RNAse P Ct values 2 standard deviations from the mean of all samples were repeated or excluded.

**Cytopathic effect (CPE) assay.** All anterior nares samples up to 14 days post symptom onset (PSO) of the index case were assayed for CPE (cytopathic effect, which is reflective of detectible infectious virus). In cases where CPE was positive within days 11–14, we continued to test samples beyond day 14 until three consecutive samples were CPE negative. CPE was assessed on Vero-hACE2-TMPRSS2 cells (gifted from A. Creanga and B. Graham at NIH) and are available from BEI Resources (NR-54970). Vero cells expressing TMPRSS2 enhanced isolation of SARS-CoV-2 compared to Vero E6 cells [48], and we reasoned that expression of hAce-2 would further enhance sensitivity to infection. Cells were maintained at 37°C and 5% $CO_2$ in Dulbecco's Modified Eagle medium (DMEM; Gibco) supplemented with 10% fetal calf serum, 100ug/mL penicillin and streptomycin (Gibco) and 10μg/mL of puromycin (Gibco). The assay was adapted from Harcourt el al. [49] and done in a 96-well format. 200uL of nasal specimen were added to a well of a 96-well plate and serially diluted 1:1 with DMEM supplemented with 1x penicillin/streptomycin over two additional wells. 100uL of freshly trypsinized cells, resuspended in infection media (made as above but with 2x penicillin/streptomycin, 5ug/mL amphotericin B [Bioworld] and no puromycin) at $2.5x10^5$ cells/mL, were added to each sample dilution. Cells were cultured at 37°C and 5% $CO_2$ and checked for CPE from day 2 to 5. Vero-hACE2-TMPRSS2 cells form characteristic syncytia upon infection with SARS-CoV-2, enabling rapid and specific visual evaluation for CPE. After 5 days of incubation, the supernatant (200uL) from one well from each dilution series was mixed 1:1 with 2x RNA/DNA

Shield (Zymo) for viral inactivation and RNA extraction as described above. Among specimens with visible CPE, the presence of infectious SARS-CoV-2 was confirmed by RT-qPCR using N primers as described above. All assays were done in the BSL3 facility at Genentech Hall, UCSF, following the study protocol that had received Biosafety Use Authorization.

**Determination of infectious viral titres.** Viral titers were determined by conventional plaque assay. Briefly, nasal specimens were diluted at 10-fold or 4-fold serial dilutions six times in DMEM (Gibco) supplemented with 100ug/mL penicillin and streptomycin (Gibco). 250uL of each sample dilutions were added the wells of 6-well plate seeded with confluent Vero-hAce2-TMPRSS2. Cells were cultured for 1hr in a humidified incubator at 37˚C in 5% $CO_2$. After incubation, 3 mL of a mixture of MEM containing a final concentration of 2% FCS, 1x penicillin-streptomycin-glutamine and 1% melted agarose, maintained at 56˚C, was added to the wells. After 72 h of culture as above, the wells were fixed with 4% paraformaldehyde for 2 hrs, agarose plugs were removed, and wells were stained with 0.1% crystal violet solution. Plaques were counted and titres were expressed as plaque forming units (pfu)/mL.

**Sequencing.** For each participant, the nasal specimen with the highest detectable RNA level, as previously determined by RT-qPCR targeting the N gene, was selected for sequencing. Whole genome sequencing was done by following the ARTIC Network amplicon-based sequencing protocol for SARS-CoV-2 [50]. Briefly, specimens were thawed and converted to cDNA using the Luna RT mix (NEB). Arctic V3 multiplex PCR primer pools (IDT) were used to generate amplicons that were barcoded using the Native Barcode expansion kits 1–24 (Nanopore), pooled and used for adaptor ligation. Libraries were run on a MinION sequencer (Oxford Nanopore Technologies) for 12–16 hours. Consensus sequences were generated using the nCoV-2019 novel coronavirus bioinformatics protocol using the MinIon Pipeline.(50) Lineage determination was done using the online Pangolin COVID-19 Lineage Assigner.

**Statistical analysis.** Baseline characteristics were summarized using medians and interquartile ranges or counts and frequencies. Symptom onset (day 0) was defined as either the first day of reported symptoms or, for those who did not report symptoms, the day of first positive SARS-CoV-2 RT-qPCR. RT-qPCR viral culture data were used to define viral shedding outcomes for each participant longitudinally. Maximum viral RNA load was defined as the highest RNA level observed among the available longitudinal specimens for each participant. Viral RNA shedding duration was defined as the last day after symptom onset in which both N and E gene targets were detected. Infectious virus shedding duration was defined as the last day after symptom onset in which CPE was observed in viral culture. Continuous variables were compared between fully vaccinated and unvaccinated participant groups using Mann-Whitney U testing and categorical variables were compared using Fisher's Exact tests. The independent effect of vaccination on viral RNA shedding was determined using repeated measures, mixed effects linear regression. For each participant, the quantity of viral RNA for each RT-qPCR target was analyzed from the day of maximum viral RNA over a seven-day period and used to predict mean viral RNA load in vaccinated and unvaccinated groups over time. Linear slopes on a log10 scale were fitted to assess interactions and estimate the difference in decline from maximum viral RNA load. Fixed effects included vaccination status (fully vaccinated or unvaccinated), time post-viral peak (measured in days post-peak viral RNA load), and the interaction between vaccination status and time; individual participants were random effects. We calculated the daily proportion of participants shedding infectious virus in each group. Because the highest proportion of vaccinated participants with infectious viral shedding occurred on day 5 after symptom onset, we created two bins (days 1–5, days 6–12). A logistic regression model, accounting for clustering by participant, was used to estimate the odds of infectious viral shedding among the groups. P-values <0.05 were considered statistically

significant. All analyses were performed using STATA/IC 16.1 (STATA Corporation, College Station, Texas, USA) and R version 3.3.2 (R Project for Statistical Computing, Vienna, Austria)

## Supporting information

**S1 Fig. Individual level shedding dynamics.** Scatter plots of each participant indicating the magnitude of viral RNA over time (days post symptom onset) and the period in which viral culture-positive samples were detected (yellow shaded area). Copies of N (grey) and E (orange) RNA targets are shown as dots in unvaccinated (A) and fully vaccinated (B) participants.
(TIF)

**S2 Fig. Effect of different sample storage conditions and freeze thaw on RNA and infectious virus quantification.** Nasal swabs collected from SARS-CoV-2 negative volunteers were added to viral transport media and spiked with SARS-CoV-2 WA1 strain or not spiked (NC). A) Copies of SARS-CoV-2 N were quantified by RT-qPCR following storage at room temperature (RT) and at 4°C for the indicated days and B) infectious titers were quantified by plaque assay following storage as indicated or following freeze-thaw (FT) cycles.
(TIF)

**S3 Fig. Correlation of RT-qPCR N and E targets.** Correlation of the magnitude of N and E targets indicating the Pearson's correlation coefficient.
(TIF)

**S4 Fig. Correlation between duration of infectious viral shedding and maximum RNA load.** Correlation between the duration of infectious viral shedding and maximum RNA load showing the Pearson's correlation coefficient.
(TIF)

**S1 Table. Oligonucleotide sequences for RT-qPCR**
(DOCX)

**S2 Table. Effect of vaccination type and time since vaccination on duration of viral shedding.**
(DOCX)

## Acknowledgments

We thank the participants for making this study possible while acutely infected with SARS-CoV-2. We appreciate the input and support of Thomas M. Lietman, Will Brett, Eric Talbert, Will Brannen, Theresa Brady, Natalie Thornburg, Ian Plumb, Holly Biggs, and others in the CDC COVID-19 response who contributed to this study. Vero TMPRSS2 hAce2 cells were a kind gift from Barney Graham (NIH). The findings and conclusions in this report are those of the author(s) and do not necessarily represent the official position of the Centers for Disease Control and Prevention (CDC).

## Author Contributions

**Conceptualization:** Miguel Garcia-Knight, Khamal Anglin, Michel Tassetto, Claire M. Midgley, Jeffrey N. Martin, Raul Andino, J. Daniel Kelly.

**Data curation:** Miguel Garcia-Knight, Khamal Anglin, Michel Tassetto, Scott Lu, Amethyst Zhang, Sarah A. Goldberg, Adam Catching, Michelle C. Davidson, Joshua R. Shak, Raul Andino, J. Daniel Kelly.

**Formal analysis:** Miguel Garcia-Knight, Khamal Anglin, Michel Tassetto, Scott Lu, Raul Andino, J. Daniel Kelly.

**Funding acquisition:** Raul Andino, J. Daniel Kelly.

**Investigation:** Miguel Garcia-Knight, Michel Tassetto, Mariela Romero, Jesus Pineda-Ramirez, Ruth Diaz-Sanchez, Paulina Rugart, Kevin Donohue, Jonathan Massachi, Hannah M. Sans, Manuella Djomaleu, Sujata Mathur, Venice Servellita, David McIlwain, Brice Gaudiliere, Jessica Chen, Enrique O. Martinez, Jacqueline M. Tavs, Charles Chiu, J. Daniel Kelly.

**Methodology:** Miguel Garcia-Knight, Khamal Anglin, Michel Tassetto, Scott Lu, Joshua R. Shak, Jesus Pineda-Ramirez, Ruth Diaz-Sanchez, Paulina Rugart, Kevin Donohue, Jonathan Massachi, Hannah M. Sans, Manuella Djomaleu, Sujata Mathur, Venice Servellita, David McIlwain, Brice Gaudiliere, Jessica Chen, Enrique O. Martinez, Jacqueline M. Tavs.

**Project administration:** Michelle C. Davidson, Melissa Briggs-Hagen, Sharon Saydah, Claire M. Midgley, Jeffrey N. Martin, Raul Andino, J. Daniel Kelly.

**Resources:** Grace Bronstone, Jacob Weiss, John T. Watson, Melissa Briggs-Hagen, Glen R. Abedi, George W. Rutherford, Steven G. Deeks, Charles Chiu, Sharon Saydah, Michael J. Peluso.

**Supervision:** Raul Andino, J. Daniel Kelly.

**Visualization:** Adam Catching.

**Writing – original draft:** Miguel Garcia-Knight, Khamal Anglin, Michel Tassetto, Scott Lu, Raul Andino, J. Daniel Kelly.

**Writing – review & editing:** Miguel Garcia-Knight, Khamal Anglin, Michel Tassetto, David McIlwain, Raul Andino, J. Daniel Kelly.

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
