## [Decision Letter · Decision Letter 0]

12 Jul 2022

Dear Dr. Andino,

Thank you very much for submitting your manuscript "Infectious viral shedding of SARS-CoV-2 Delta following vaccination: a longitudinal cohort study" for consideration at PLOS Pathogens. As with all papers reviewed by the journal, your manuscript was reviewed by members of the editorial board and by several independent reviewers. In light of the reviews (below this email), we would like to invite the resubmission of a revised version that takes into account the reviewers' comments.

All reviewers concurred that the study was well-done and that the inclusion of infectious virus detection (in addition to viral RNA load) is a particular strength. Opinions differed on the novelty of the work as different VOC have come to dominate since this study was completed. However, the data appears rigorous and exemplifies the kind of analysis which might be useful to the field. Nonetheless, all reviewers provided constructive comments on how to better qualify the conclusions made and also include better references to the literature in this fast moving field.  

We cannot make any decision about publication until we have seen the revised manuscript and your response to the reviewers' comments. I urge the authors to answer the reviewers' comments as clearly and as comprehensively as they can.  Your revised manuscript may be sent to back to reviewers for further evaluation if I the revisions are less than comprehensive.  

Sincerely,

Benhur Lee

Section Editor

PLOS Pathogens

Benhur Lee

Section Editor

PLOS Pathogens

Kasturi Haldar

Editor-in-Chief

PLOS Pathogens

orcid.org/0000-0001-5065-158X

Michael Malim

Editor-in-Chief

PLOS Pathogens

orcid.org/0000-0002-7699-2064

Reviewer's Responses to Questions

**Part I - Summary**

Reviewer #1: Garcia-Knight et al show how viral shedding of SARS-CoV-2 compares in unvaccinated and vaccinated individuals. The authors used a rigorous testing regimen to compare daily RNA levels and viral shedding from the same individuals. Based on their data, they suggest that although peak viral levels did not differ according to vaccination status, vaccinated individuals cleared infectious virus earlier than unvaccinated individuals. The strength of the study is the direct comparison of viral RNA levels with infectious virus over a pro-longed time period (> 10 days post symptom-onset) from over 80 individuals. One main weakness of the study is that unvaccinated and vaccinated individuals where infected by different virus populations which warrants caution when interpreting the data especially if one is to project this on the current situation with Omicron and booster roll-out. Nevertheless, with this manuscript, the authors present valuable information for evaluating testing and isolation/quarantine recommendations as well as the impact of vaccination programs on SARS-CoV-2 transmission. The study is well organized and the data justify the conclusions by the authors which are of relevance to inform public health interventions to control SARS-CoV-2 onward transmission.

Only minor comments as listed below might be considered to improve the manuscript.

Reviewer #2: In this analysis, the authors present a longitudinal study of viral load dynamics of 32 vaccinated and 52 unvaccinated individuals within the first 5 days after symptom onset, which demonstrated that although the peak viral load were similar between both groups, vaccinated individuals had a faster viral load and infectious virus decline and were less likely to have infectious virus post 5 days after symptom onset. The analysis is carefully carried out, and the results are nicely presented. I have a few observations and suggestions to make.

Reviewer #3: The authors report the impact of vaccination on infectious viral shedding of SARS-CoV-2 before and during the Delta wave. The study is based on longitudinal nasal swabs from a relatively small cohort. The inclusion of infectious virus quantification in addition to viral RNA quantification is important to draw any conclusion about infectiousness.

The overall experimental approach is sound and well executed, and the results are reported clearly.

The novelty of the work however is of concern; similar studies have already been published regarding the delta variant and that variant has now been replaced by omicron and its sublineages rendering the conclusions less impactful in terms of public health guidance.

**Part II – Major Issues: Key Experiments Required for Acceptance**

Reviewer #1: NA

Reviewer #2: 1- It would be very useful for authors to present clearly how many proportionally in the vaccinated vs unvaccinated had infectious virus post day 5 and day 10. This would help understand how many individuals are still potentially infectious. A recent study https://www.ncbi.nlm.nih.gov/pmc/articles/PMC8996632/ has shown that only 17% had a culture-positive virus post day 5.

2- There is also another confounder that was not addressed here, whether the unvaccinated had a previous infection. Although the effect of prior infection on infectiousness is not well studied, the immune response may suppress the infectious virus nevertheless. This should be noted as a limitation if this information is not available or included in the analysis.

3- Would it be possible to also look at the infectiousness duration among those vaccinated before 3 months vs after 3 months. Singanayagam et al. show that some of the vaccine's protective effects may start waning after 2-3 months, which may be why we are observing different results in different studies. The authors need to add this recently published paper: https://www.nejm.org/doi/full/10.1056/NEJMc2202092 showing that most patients have culture-positive virus post day 5. But the difference between NEJM paper and CID paper https://www.ncbi.nlm.nih.gov/pmc/articles/PMC8996632/ and current analyses may be to do with host factors (age, comorbidities etc.) as well as specimen collection time, time from vaccination, and variant. This would be a good discussion point, in my view.

Reviewer #3: It would be beneficial to include the results of the experiments comparing the effect of freeze-thaw cycles on viral RNA and infectious virus quantification.

**Part III – Minor Issues: Editorial and Data Presentation Modifications**

Reviewer #1: 1. Self-reported symptom onset can underly several biases e.g. recall bias. Thus, the authors might include some discussion on whether time of symptom onset is comparable between the groups infected with different viruses or vaccinated / unvaccinated individuals. E.g. Puhach et al 2022 shows that pre-VOC SARS-CoV-2 infections had lower infectious viral loads than Delta-infected unvaccinated individuals during the first 5 symptomatic days, which might also influence symptom onset.

2. A direct comparison of viral kinetics and decay in Delta unvaccinated individuals would be preferable, however the study here focuses on the potential of onward transmission from vaccinated individuals in general. The findings presented here underline that vaccination reduced the likelihood of onward transmission from day 5 onwards, although from data presented in Figure 3B it looks like this also holds true for days 1-5? Figure 3 should be described in more detail in the figure legend. For Figure 3B, please elaborate on what exactly was compared and how the shown p-values were derived. It looks like for all days up to day 5 the proportion of viral culture + samples in the unvaccinated is higher than for the vaccinated, thus it is not clear whether the shown p-value refers to a comparison between d1 & day 5 among the fully vaccinated or comparing unvaccinated vs vaccinated overall. An improved image quality should be handed in for final submission.

4. It should be underlined that even as infectious virus can be isolated from patient’s samples there is no defined cut-off value for virus concentrations that would lead to actual transmission. Thus, the authors might want to include some guidance on how their findings would influence quarantine recommendations. However, these implications might be not directly transferable to the post-booster era especially in the context of emerging Omicron variants.

6. Update Bibliography e.g. Puhach et al 2022 is now published in Nature Medicine

Reviewer #2: 1- Although detecting the infectious virus is important and somewhat can be used as an indicator to understand the infectiousness potential of the host, there is now evidence that viral load is also important in transmission. Detecting culture-positive virus late in infection when viral load is very low may not lead to transmission. So, I believe some of the comments in the discussion need to be softened to address this. This paper showed that peak viral load happens early in infection https://pubmed.ncbi.nlm.nih.gov/33521734/, likely contributing to most of the transmission events. I am yet to see a report demonstrating onward transmission late in infection. For instance, this study https://pubmed.ncbi.nlm.nih.gov/33530000/ showed that those with persistent RNA positivity had zero onward transmission to their contacts. Therefore, an infectious virus should not be equated to infectiousness as it depends on many factors. Also, important to address that the presence of an immune response will suppress the viable virus. So, detecting culture-positive virus in a vaccinated population may not be the same as detecting culture positive in an unvaccinated and naïve population.

2- One of the caveats of the study, as also addressed briefly by the authors, is that the patients were included within 5 days after symptom onset. as they presented, 31% of unvaccinated individuals had very high Ct values, meaning they were already beyond the peak viral load. But, also there is a caveat that delta infections likely had an earlier peak so this study might have missed the peak in those individuals too. So, studies like this one https://pubmed.ncbi.nlm.nih.gov/34756186/ where contacts are recruited and followed up, will provide much better information about the peak viral load. So, I think the results of this study should be highlighted– the results are in line with the current analysis. However, they haven't investigated infectious virus; they had epidemiological evidence of onward transmission: "The mean viral load decline rate of the fully vaccinated delta group was also faster than those of the alpha group (pp=0·84) and the unvaccinated delta group (pp=0·85)."

3- I agree with the authors' conclusion that their analysis underestimates the true difference in infectiousness period between vaccinated and unvaccinated.

4- It's unclear why the authors used Vero-hACE2-TMPRSS2 cells for cell lines; more information is needed to support this approach. Was there a specific reason or rationale? It would be good to provide some background information about using these cell lines.

Reviewer #3: This paper would greatly benefit from an update literature search about longitudinal studies of infectious viral shedding. Several studies have been published that do not appear in the the references. Also, some of the referenced pre-prints might now be published.

PLOS authors have the option to publish the peer review history of their article (what does this mean?). If published, this will include your full peer review and any attached files.

Reviewer #1: No

Reviewer #2: No

Reviewer #3: No
---

## [Editor Report · Decision Letter 1]

5 Aug 2022

Dear Dr. Andino,

We are pleased to inform you that your manuscript 'Infectious viral shedding of SARS-CoV-2 Delta following vaccination: a longitudinal cohort study' has been provisionally accepted for publication in PLOS Pathogens.

Best regards,

Benhur Lee

Section Editor

PLOS Pathogens

Benhur Lee

Section Editor

PLOS Pathogens

Kasturi Haldar

Editor-in-Chief

PLOS Pathogens

orcid.org/0000-0001-5065-158X

Michael Malim

Editor-in-Chief

PLOS Pathogens

orcid.org/0000-0002-7699-2064
---

## [Editor Report · Acceptance letter]

7 Sep 2022

Dear Dr. Andino,

We are delighted to inform you that your manuscript, "Infectious viral shedding of SARS-CoV-2 Delta following vaccination: a longitudinal cohort study," has been formally accepted for publication in PLOS Pathogens.

Best regards,

Kasturi Haldar

Editor-in-Chief

PLOS Pathogens

orcid.org/0000-0001-5065-158X

Michael Malim

Editor-in-Chief

PLOS Pathogens

orcid.org/0000-0002-7699-2064